# Agri-Ecological Policy, Human Capital and Agricultural Green Technology Progress

**Guoqun Ma** [1,2], **Minjuan Li** [1], **Yuxi Luo** [1,2,3] **and Tuanbiao Jiang** [1,2,4,*]

1.  School of Economics and Management, Guangxi Normal University, Guilin 541006, China; yluogxnu@mailbox.gxnu.edu.cn (Y.L.)
2.  Pearl River-Xijiang River Economic Belt Development Institute, Guangxi Normal University, Guilin 541004, China
3.  Guangxi Key Laboratory of Landscape Resources Conservation and Sustainable Utilization in Lijiang River Basin, Guangxi Normal University, Guilin 541004, China
4.  Center for Southwest Urban and Regional Development, Guangxi Normal University, Guilin 541004, China
*   Correspondence: 452697151@mailbox.gxnu.edu.cn

**Abstract:** Agri-ecological policy and human capital are important factors affecting agricultural green technology progress (AGTP), but the related research is relatively insufficient. This paper contributes to existing research through new insights on the effect of agri-ecological policy on AGTP, using human capital as a moderating variable. We use the Super-efficiency SBM-DEA model to measure AGTP in 30 provinces of China during 2000–2019, and use a two-way fixed effects model to analyze the nonlinear effect of agri-ecological policy on AGTP and the moderating role of human capital. The results show that there is a "U"-shaped relationship between agri-ecological policy and AGTP, where human capital plays a positive moderating role. Intermediate human capital and advanced human capital can significantly moderate the impact of agri-ecological policy on AGTP, while the moderating role of primary human capital is not significant. The "U"-shaped relationship between agri-ecological policy and AGTP involves some heterogeneity based on differences in grain function and the two sides of the Hu Huanyong line.

**Keywords:** agri-ecological policy; human capital; agricultural green technology progress; panel data model

## 1. Introduction

Green development of agriculture has become the core concern in China's agricultural development [1]. However, China's agriculture still suffers from serious ecological problems such as excessive use of chemical production factors, excessive consumption of cultivated land and destruction of agricultural produce [2]. In order to realize the green development of agriculture, it is necessary to guarantee agricultural output while reducing agricultural pollution emissions and protecting the agricultural environment [3]. Therefore, the Chinese government has introduced a series of agri-ecological policy measures. Theoretically, the implementation of agri-ecological policy can force agricultural producers to engage with technology innovation, and provide policy drivers with a green and low-carbon development of agriculture [4]. However, agri-ecological policy also constrains agricultural production behavior, increases the cost of agricultural technology research and adversely affects agricultural green technology progress [5].

The effectiveness of an agri-ecological policy can also be influenced by the human capital of agricultural producers [6]. Agricultural producers with high human capital also have high innovation ability, and they will actively conduct research and development on agricultural green technology while facing the constraints of agri-ecological policy [7]. However, the increase of human capital will promote the outflow of agricultural labor to non-agricultural industries, hindering the agricultural green technology progress [8]. So,

what is the impact of agri-ecological policy on agricultural green technology progress? What is the role of human capital in the relationship between agri-ecological policy and agricultural green technology progress? These are questions that deserve further exploration.

Previous studies have explored the effect of agri-ecological policy and agriculture green technology progress (AGTP), finding that agri-ecological policy is an important factor affecting AGTP [9]. Some scholars found that agri-ecological policy measures such as soil health cards and soil formula fertilization can accelerate agriculture green technology progress by improving soil fertility [10] and enhancing the profitability of farming [11]. However, there is still no consensus regarding the effect of agri-ecological policy on AGTP [12]. Some scholars point out that agri-ecological policy can improve agricultural producers' knowledge and innovation level, thus promoting AGTP [13]. Other scholars believe that agri-ecological policy have an adverse effect on the replacement of chemical fertilizer with organic manure by agricultural producers. While incentive-based agri-ecological policy can promote the replacement of chemical fertilizer with organic manure [14], it will lead to the excessive use of fertilizer and inhibit AGTP [15]. The divergence of existing studies on the effect of agri-ecological policy is due to the dynamic changes in agri-ecological policy being neglected [16]. In the early stage of agri-ecological policy, agricultural producers will invest part of their agricultural funds in agricultural environmental pollution control to meet policy requirements. This will constrain the input of agricultural technology research, inhibiting AGTP [17]. With the implementation of agri-ecological policy and the application of green production technology, agricultural pollutants will decrease, and the agricultural products produced by green production technology will have higher added value and market prices, which will also increase the profits of agricultural producers [18]. This will in turn promote AGTP [19].

Existing studies claim that human capital can have an effect on AGTP [20,21]. However, few studies have explored the role of human capital in the relationship between agri-ecological policy and AGTP [16]. Indeed, the human capital of agricultural producers can directly influence the effect of agri-ecological policy [22]. Agricultural producers with higher human capital will tend to adopt agricultural green production technology to meet the requirements of environmental supervision, which can partly offset the negative effect of agri-ecological policy [23]. However, agricultural producers with higher human capital will be more inclined to shift to non-agricultural industry, leading to a lack of highly qualified agricultural labor and hindering AGTP [17].

This paper makes a contribution to the existing literature from the following three perspectives. First, we provide a nonlinear discussion about the relationship between agri-ecological policy and AGTP. Previous studies on agri-ecological policy and AGTP mainly explored the static effect, ignoring the dynamic impact of agri-ecological policy [24]. We investigate the nonlinear effect of agri-ecological policy on AGTP by adding the square term of agri-ecological policy. Second, the existing literature has explored the relationship between agri-ecological policy and AGTP, but the role played by the object of regulation (i.e., agricultural producers) has not been examined. We examine the role of the human capital of agricultural producers in agri-ecological policy [25]. Finally, this paper fully considers the heterogeneity of agri-ecological policy [26]. Various regions have different natural resource endowments, and we try to explore the differential effects of agri-ecological policy under heterogeneous conditions. We hope to evaluate the impact of China's agri-ecological policy on AGTP and the role of human capital in it through research on the above three issues.

In this study, we provide empirical evidence to support the notions that agri-ecological policy has a significant effect on agricultural green technology progress, and that human capital plays a moderating role in the effect of agri-ecological policy on agricultural green technology progress.

## 2. Theoretical Analysis and Research Hypotheses

*2.1. The impact of Agri-Ecological Policy on the Agricultural Green Technology Progress*

Environmental pollution is a typical public goods problem, and the solution to this problem relies heavily on agri-ecological policy [27]. At the initial stage of the implementation of agri-ecological policy measures, agricultural producers have to reduce the use of chemical production factors to meet policy requirements, which will reduce crop yields and reduce the profitability of agricultural producers [14–28]. At the same time, the implementation of agri-ecological policy will also raise the production costs of pesticides and fertilizers [29]. Given their profit orientation, the manufacturers of pesticides and fertilizers will transfer the rising costs to agricultural producers. This will reduce the inclination of agricultural producers to adopt green production technology, negatively affecting agricultural green technology progress [28]. In addition, due to the low intensity of agri-ecological policy and the low binding force of policies, the traditional production methods oriented to maximize agricultural output may not change, which also invites problems such as the stagnation of agricultural green technology [17].

With the increase of policy intensity, agri-ecological policy measures will become more stringent, which will force agricultural producers to make a green transition [30]. On the one hand, agricultural producers' awareness of environmental protection will increase. They will be more willing to develop and apply agricultural green production technology, which will promote AGTP [31]. In addition, agri-ecological policy can also have an impact on AGTP by enhancing farmers' technical interactivity. This is because agricultural producers' research on and application of agricultural green production technology will gradually increase, which can improve profitability for those engaged in green production. Due to the demonstrative nature of agricultural production and technology application, other agricultural producers will also engage in agricultural green production through learning, imitation and other methods, which can further accelerate agricultural green technology progress [32]. On the other hand, financial subsidies in agri-ecological policy are also increasing. This will motivate agricultural producers to make the green technology transition [33]. At the same time, the strengthening of some binding agri-ecological policy will also force agricultural producers to reduce the use of chemical products. This will reduce agricultural pollution and promote AGTP [34]. The above analysis indicates that the effect of agri-ecological policy on AGTP will exhibit nonlinear characteristics. Thus, this paper proposes Hypothesis 1.

**H1.** *There is a "U"-shaped relationship between agri-ecological policy and agricultural green technology progress.*

*2.2. The Moderating Role of Human Capital between Agri-Ecological Policy and Agricultural Green Technology Progress*

China's agricultural production can be characterized as a smallholder economy [35]. The human capital of agricultural producers also determines the effectiveness of agri-ecological policy [6]. Firstly, under the influence of agri-ecological policy, the human capital of agricultural producers can influence AGTP through resource allocation [36]. Specifically, agricultural producers with higher human capital also have stronger environmental awareness, and they will pay more attention to the environmental pollution problems caused by agricultural production consciously, comply with policies related to agri-ecological policy, and choose to employ green production factors [37]. Secondly, agricultural producers with higher human capital are more likely to realize that the quality premium of agricultural products brought by green production methods is higher than the loss of production profit due to the reduction of chemical factors [38]. Thus, they tend to choose agricultural production methods based on agricultural green technology, which in turn promotes AGTP [39]. In addition, the human capital of agricultural producers is closely related to their cultural level, and agricultural producers with higher cultural levels are more aware of the damage caused by the excessive use of chemical elements and farmland to the agricultural ecologi-

cal environment. Therefore, it is also easier for them to accept the relevant regulations on environmental protection and pollution prevention in agri-ecological policy, which makes it easier to develop and apply agricultural green production technology [40].

Thirdly, the higher the agri-ecological policy intensity, the higher the degree of environmental concern of agricultural producers with higher human capital will be. Their willingness and ability to adopt green production technology will also be stronger [41]. Meanwhile, under the influence of agri-ecological policy, in order to reduce production costs and improve the quality of agricultural products, agricultural producers with higher human capital will also actively optimize agricultural production technology and promote the upgrading of agricultural production methods [42]. Finally, the siphoning off of human capital also affects the relationship between agri-ecological policy and AGTP [20]. The higher the human capital of agricultural producers in a region, the faster the solution of environmental problems and the improvement of green technology innovation [43]. The higher agricultural green technology progress will attract agricultural producers with high human capital in other regions through the spillover effect, which will further enhance the human capital of agricultural producers in the region [44]. Thus, this paper proposes Hypothesis 2.

**H2.** *Human capital can moderate the relationship between agri-ecological policy and agricultural green technology progress.*

### 3. Empirical Model and Methodology

*3.1. Empirical Model*

In order to examine the non-linear impact of agri-ecological policy on the agricultural green technology progress (Hypothesis 1), this paper introduces the squared term of agri-ecological policy and constructs an econometric model (1) as follows [45]:

$$AGTP_{it} = \alpha_0 + \alpha_1 AEP_{it} + \alpha_2 (AEP_{it})^2 + \sum_j \alpha_j X_{it} + \delta_i + \theta_t + \mu_{it} \tag{1}$$

where *AGTP* represents agricultural green technology progress, *AEP* represents agri-ecological policy, *X* indicates a series of control variables, $\delta_i$ and $\theta_t$ are individual-fixed effects and time-fixed effects, respectively, and $\mu_{it}$ denotes the random disturbance term. Meanwhile, in order to further test the moderating role of human capital, this paper refers to existing studies [46], adding the interaction terms of human capital and agri-ecological policy's linear term and its squared term to construct empirical model (2).

$$AGTP_{it} = \beta_0 + \beta_1 AEP_{it} + \beta_2 (AEP_{it})^2 + \beta_3 EDU_{it} + \beta_4 EDU_{it} * AEP_{it} \\ + \beta_5 EDU_{it} * (AEP_{it})^2 + \sum_j \beta_j X_{it} + \delta_i + \theta_t + \varepsilon_{it} \tag{2}$$

Model (2) is used to test Hypothesis 2. Here, *EDU* denotes human capital, $\varepsilon_{it}$ denotes the random disturbance term, the other variables are same as Equation (1). If $\beta_4$ is greater than 0, this indicates that human capital plays a positive moderating role in the process of agri-ecological policy on *AGTP*, and vice versa. If $\beta_5$ is less than 0, this means that the "U" curve between agri-ecological policy and *AGTP* becomes flat under the influence of human capital. That is, human capital can reduce the negative effect of agri-ecological policy on *AGTP*, and vice versa.

*3.2. Variable Selection*

3.2.1. The Explained Variable

Agricultural Green Technology Progress (*AGTP*): This is a kind of agricultural green technology efficiency based on technology progress [47], which usually refers to technology innovation from the perspective of combining economic performance and environmental issues [48]. Previous studies used the Solow residual method, the stochastic frontier production function method (SFA) and data envelopment analysis (DEA) to measure agricultural

green technology progress [49,50]. Consider that the assumptions of perfect competition and constant returns to scale in the Solow residual method do not conform to the reality of China's agricultural development [51,52]. For SFA, researchers need to set the production function form subjectively, and the price information of China's agricultural pollutant emissions is needed, which is unavailable [53]. Therefore, this paper uses the DEA method to measure China's agricultural green technology progress [54]. Meanwhile, considering the Super-efficiency SBM-DEA model, one can avoid the measurement error of a subjective setting production function, avoid the overestimation of technical efficiency when there is nonzero relaxation in the input or output, and sort the effective decision-making units [49]. We use the Super-efficiency SBM-DEA Model to measure the *AGTP* of 30 provinces in China with reference to [21]. The selected variables and the data source are shown in Table 1. Among them, the agricultural non-point source pollution emissions include chemical oxygen demand, total nitrogen and total phosphorus in the water, pesticide residues and agricultural film residues in the soil, with calculation methods as in [55]. The carbon emissions mainly consist of carbon dioxide released by various agricultural production activities, with calculation methods as in [56]. Considering the proportion requirement of the DEA model on input and output, we combined agricultural non-point source pollution emissions into an agricultural pollution comprehensive index with reference to [57]. In addition, we converted *AGTP* into a fixed-base index to reflect its cumulative change characteristic. That is, we assigned the value of *AGTP* in 2000 to 1; then, *AGTP* in 2001 is the product of the actual value of *AGTP* in that year and the value in 2000, *AGTP* in 2002 is the product of the actual value of *AGTP* in that year and the value in 2001 and the value in 2000, and so on.

**Table 1.** The measurement indicators of *AGTP*.

| | | | |
|---|---|---|---|
| **Input Indicators** | Labor | Number of labor force in planting industry | China Statistical Yearbook |
| | Land | Total crop area sown | China Rural Statistical Yearbook |
| | Livestock | Number of large livestock | China Rural Statistical Yearbook |
| | Mechanical power | Total power of agricultural machinery | China Rural Statistical Yearbook |
| | Irrigation | Actual irrigated area | China Rural Statistical Yearbook |
| | Pesticides | Number of pesticides used | China Rural Statistical Yearbook |
| | Agricultural film | Total weight of agricultural films used | China Rural Statistical Yearbook |
| | Fertilizer input | Total weight of chemical fertilizer application | China Rural Statistical Yearbook |
| **Output Indicators** | Desirable Output | Total agricultural output value at constant prices with 2000 as the base period | China Statistical Yearbook |
| | Undesirable Output | Agricultural pollution comprehensive index and carbon emissions | Calculated by the author |

### 3.2.2. The Explanatory Variable

Agri-ecological policy (*AEP*): This refers to the government's policies to regulate the production behavior of agricultural producers through guidance, incentives, and constraints. Some scholars used environmental pollution levels, abatement costs, or actual levy costs to measure agri-ecological policy from an ex-post perspective [58]. However, the above indicators are all the results of agricultural green technology progress, and the use of these indicators will lead to endogeneity issues [49]. Therefore, this paper uses the quantity of ecological policies as the measure of agri-ecological policy based on existing research from an ex-ante perspective [59]. The number of agri-ecological policy reflects the government's concern with environmental protection; the higher the number of agri-ecological policy, the higher the government's attention to agricultural ecological environment protection. Considering that after 2000 the Chinese government has attached great importance to agricultural environmental governance, we refer to existing studies [56,57] using the number of policies related to environmental protection and pollution prevention in the agriculture sector as the instrumental variable and transforming by taking the logarithm.

### 3.2.3. The Moderating Variable

Human capital (*EDU*): Farmers are the main decision makers of agricultural production, the dischargers of agricultural pollution and the implementers of agricultural pollution control [60]. Therefore, the human capital of agricultural producers not only determines their perceptions of agri-ecological policy, but also influences their adoption and innovation of agricultural green technology [61]. Previous studies used the income method, the expenditure method and the education indicator method to measure human capital [62]. Since China's farmers pay more attention to education investment compared with other investment expenditures [63], we refer to the existing literature and use the average years of schooling to measure human capital [56]. In order to further investigate the moderating effect of heterogeneous human capital between agri-ecological policy and *AGTP*, we classified human capital into primary human capital (*EDU1*), intermediate human capital (*EDU2*), and advanced human capital (*EDU3*) according to education level, with reference to [64].

### 3.2.4. The Control Variables

Referring to [65,66], we selected the following control variables: (1) Trade dependence (*TRA*), expressed as the ratio of total import and export of agricultural products to gross agricultural product; (2) the density of agricultural machinery (*MAC*), expressed as the total power of agricultural machinery per unit of sown area; (3) urbanization (*URB*), expressed as the proportion of urban population to the total population; (4) the degree of disaster exposure (*ADR*), expressed as the share of disaster area to the total sown area of crops. Considering that the latest official data about human capital variables are only up to date as of 2019, the study interval of this paper is 2000–2019. The descriptive statistical characteristics and the data sources of the variables are shown in Table 2.

**Table 2.** Descriptive statistical characteristics of variables.

| Variable Name | Mean | SD | Max. | Min. | Data Source |
|---|---|---|---|---|---|
| Agricultural Green Technology Progress | 1.467 | 0.687 | 3.903 | 0.428 | Calculated by the author |
| Agri-ecological policy | 3.875 | 0.640 | 4.860 | 2.482 | China Environment Statistics Yearbook |
| Human Capital | 7.507 | 0.691 | 9.444 | 5.737 | |
| Primary Human Capital | 6.084 | 0.444 | 7.074 | 4.602 | China Population and Employment |
| Intermediate Human Capital | 1.035 | 0.406 | 2.871 | 0.270 | Statistics Yearbook |
| Advanced Human Capital | 0.386 | 0.335 | 2.470 | 0.022 | |
| Trade Dependence | 0.304 | 0.368 | 1.567 | 0.034 | China Agricultural Yearbook |
| Degree of disaster exposure | 0.228 | 0.154 | 0.695 | 0.015 | China Rural Statistical Yearbook |
| Density of agricultural machinery | 0.558 | 0.264 | 1.280 | 0.177 | China Rural Statistical Yearbook |
| Urbanization | 0.517 | 0.150 | 0.891 | 0.244 | China Statistical Yearbook |

## 4. Results

### 4.1. Regression Results of Agri-Ecological Policy on AGTP

We used Stata 16.0 to analyze the non-linear effect of agri-ecological policy on agricultural green technology progress through a two-way fixed effects panel data model; the results are shown in Table 3. It can be seen that the coefficient of *AEP* on agricultural green technology progress in regression (1) is negative and the *p*-value of the *t*-test is less than 1%, indicating that, in the short run, agri-ecological policy has a negative impact on agricultural green technology progress. The coefficient of $AEP^2$ is positive and the *p*-value of the *t*-test is less than 1%, indicating that the impact of agri-ecological policy on agricultural technology progress shows a "U"-shaped trend, and Hypothesis 1 is confirmed. This is because, at the initial stage of the implementation of agri-ecological policy measures, agricultural producers are forced to increase the cost of environmental pollution control to meet the corresponding policy requirements, which will override the expenditures for agricultural green technology research, making the impact of agri-ecological policy show an inhibition effect. With the increasing intensity of agri-ecological policy, some agricultural producers

begin to realize the importance of green production and the competitive advantage of green agricultural products in the market, and will take the initiative to innovate and apply green production technology, which will gradually offset the negative impact of the rising cost of agri-ecological policy and eventually improve their agricultural green technology progress.

**Table 3.** Results of agri-ecological policy on agricultural green technology progress.

| Variables | Regression (1) | | Regression (2) | |
|---|---|---|---|---|
| | Coefficient | t-Value | Coefficient | t-Value |
| $AEP^2$ | 0.158 *** | 3.52 | 1.974 *** | 2.75 |
| $AEP$ | −0.918 *** | −2.87 | −19.476 *** | −3.61 |
| $EDU$ | | | −3.047 ** | −2.51 |
| $AEP \times EDU$ | | | 1.373 ** | 2.13 |
| $AEP^2 \times EDU$ | | | −0.162 * | −1.93 |
| $TRA$ | 0.082 | 0.49 | 0.290 * | 1.72 |
| $ADR$ | −0.262 * | −1.75 | −0.165 | −1.15 |
| $MAC$ | 0.849 *** | 4.47 | 0.498 *** | 2.63 |
| $URB$ | 1.682 *** | 3.42 | 1.320 *** | 2.73 |
| $Cons\_$ | 1.274 ** | 2.07 | 41.297 *** | 4.20 |
| Individual effect | Yes | | Yes | |
| Time effect | Yes | | Yes | |
| $R^2$ | 0.530 | | 0.641 | |

Note: *** indicates $p < 0.01$, ** indicates $p < 0.05$, * indicates $p < 0.1$; t-values in parentheses.

We further examined the moderating role of human capital between agri-ecological policy and agricultural green technology progress, and the results are presented in regression (2) of Table 3. It can be seen that the influence direction of $AEP$ and $AEP^2$ are consistent with regression (1). The coefficient of $AEP \times EDU$ is positive and the coefficient of $AEP^2 \times EDU$ is negative, and the *p*-values of the *t*-test are less than 5% and 10%, respectively, indicating that human capital can smooth the "U"-shaped relationship between agri-ecological policy and agricultural green technology progress. That is, human capital can positively moderate the nonlinear relationship between agri-ecological policy and agricultural green technology progress, and Hypothesis 2 is confirmed. This is the case because agricultural producers with higher human capital often have higher cultural levels, and their willingness to accept and apply green production technology is higher, making it easier to allocate resources reasonably. This can alleviate the negative impact of agri-ecological policy on agricultural green technology progress. Meanwhile, the increase in human capital brought about by higher cultural levels can also accelerate the application speed of agricultural green technology, which can increase the added value of agricultural products and the income of agricultural producers, and in turn enhance the cultural level and environmental awareness of agricultural producers, enabling them to actively develop and apply agricultural green production technology.

*4.2. Analysis of the Moderating Effect of Heterogeneous Human Capital*

Here, we consider the differences between various kinds of human capital in the face of technological progress. We further investigated the moderating role of heterogeneous human capital (see Table 4). Regression (1) in Table 4 is the result of the moderating effect of primary human capital, and it can be seen that the coefficients of $AEP \times EDU_1$ and $AEP^2 \times EDU_1$ are both not significant, indicating that the moderate role of primary human capital is relatively weak. This is because primary human capital has relatively low education level and environmental awareness. The relevant actors may not consciously follow the requirements when agri-ecological policy causes losses to farmers' interests. In addition, the low education level of primary human capital will also hinder its mastery of green production technology.

**Table 4.** Moderating effect of heterogeneous human capital.

| Variables | Regression (1) | | Regression (2) | | Regression (3) | |
|---|---|---|---|---|---|---|
| | Coefficient | t-Value | Coefficient | t-Value | Coefficient | t-Value |
| $AEP^2$ | 0.323 | 0.61 | 0.498 *** | 4.44 | 0.346 *** | 5.29 |
| $AEP$ | −3.631 | −0.92 | −3.697 *** | −4.52 | −2.341 *** | −4.99 |
| $AEP \times EDU_i$ | 0.475 | 0.73 | 3.425 *** | 4.18 | 5.000 *** | 3.11 |
| $AEP^2 \times EDU_i$ | −0.031 | −0.35 | −0.421 *** | −3.95 | −0.654 *** | −3.44 |
| $TRA$ | 0.075 | 0.46 | 0.102 | 0.59 | 0.104 | 0.6 |
| $ADR$ | −0.22 | −1.49 | −0.275 * | −1.86 | −0.205 | −1.38 |
| $MAC$ | 0.906 *** | 4.71 | 0.918 *** | 4.85 | 0.870 *** | 4.63 |
| $URB$ | 1.723 *** | 3.54 | 1.692 *** | 3.44 | 1.523 *** | 3.13 |
| $EDU_i$ | −1.521 | −1.29 | −6.749 *** | −4.27 | −9.038 *** | −2.65 |
| Constant term | 10.135 | 1.42 | 6.693 *** | 4.42 | 3.853 *** | 4.37 |
| Individual effect | Yes | | Yes | | Yes | |
| Time effect | Yes | | Yes | | Yes | |
| $R^2$ | 0.55 | | 0.55 | | 0.55 | |

Note: i = 1, 2, and 3 denote primary human capital, intermediate human capital, and advanced human capital, respectively. *** indicates $p < 0.01$, * indicates $p < 0.1$.

Regression (2) and regression (3) account for the moderating effects of intermediate human capital and advanced human capital, respectively. It can be seen that the coefficients of $AEP \times EDU2$ and $AEP \times EDU3$ are both significantly positive, and the $p$-values of the $t$-test are both less than 1%, indicating that intermediate human capital and advanced human capital are able to positively moderate the relationship between agri-ecological policy and agricultural green technology progress. The coefficients of $AEP^2 \times EDU2$ and $AEP^2 \times EDU3$ are negative, and the $p$-values of the $t$-test are less than 1%, indicating that the "U"-shaped relationship between agri-ecological policy and agricultural green technology progress can be smoothed by intermediate human capital and advanced human capital. That is, the negative effect of agri-ecological policy on agricultural green technology progress is attenuated under the influence of intermediate human capital and advanced human capital. This is the case because agricultural producers with intermediate human capital and advanced human capital have a more environmentally conscious and innovative spirit. They are more inclined to use agricultural green technology innovations and agricultural green production technology. This will increase the added value of agricultural products and the profit of agricultural producers, partly offsetting the negative effects of agri-ecological policy.

*4.3. Robustness Tests*

We used the following methods to conduct robustness tests. Firstly, we refer to [67], using the amount of regional industrial pollution control investment to GDP as the alternative core explanatory variables, and the regression result is shown in regression (1) of Table 5. Secondly, considering the development gap between municipalities directly under the central government and the provinces, we excluded the data of four municipalities directly under the central government to avoid sample errors, and the regression result is shown in regression (2) of Table 5. Thirdly, considering the "No. 1 Document" of the central government focus on agricultural pollution prevention and ecological restoration since 2013, which has promoted the green development of agriculture, we excluded data from before 2013, and the regression result is shown in regression (3) of Table 5. Finally, considering that the implementation of agri-ecological policy often involves hysteresis, we lag the core explanatory variables by one period and used it for regression analysis; the result is shown in regression (4) of Table 5. It can be seen that all the regression results of the robustness test are generally consistent with the regression results in Table 4, indicating that the regression results in Table 4 have good robustness.

**Table 5.** Robustness tests of agri-ecological policy on agricultural green technology progress.

| Variables | Regression (1) | Regression (2) | Regression (3) | Regression (4) |
|---|---|---|---|---|
| $AEP^2$ | 0.168 *** | 0.159 *** | 2.587 *** | |
| | (3.63) | (3.11) | (4.56) | |
| $AEP$ | −0.977 *** | −1.086 *** | −23.584 *** | |
| | (−2.95) | (−2.98) | (−4.61) | |
| $L.\,AEP^2$ | | | | 0.140 *** |
| | | | | (2.91) |
| $L.\,AEP$ | | | | −0.692 ** |
| | | | | (−2.04) |
| Constant term | 1.389 ** | 1.397 ** | 52.063 *** | 0.831 |
| | (2.19) | (2.00) | (4.49) | (1.29) |
| Control variable | Yes | Yes | Yes | Yes |
| Individual effect | Yes | Yes | Yes | Yes |
| Time effect | Yes | Yes | Yes | Yes |
| $R^2$ | 0.515 | 0.531 | 0.233 | 0.536 |

Note: *** indicates $p < 0.01$, ** indicates $p < 0.05$; the numbers in parentheses is the t-value.

### 4.4. Regional Heterogeneity Test

Functional food production zones have ensured the supply of China's agricultural products. However, with accelerated urbanization, the environmental conditions of various functional grain production areas have been deteriorating, which brings with it challenges for the agricultural green technology progress. Therefore, based on the classification criteria for functional grain production areas in the National Medium & Long-term Plan for Food Security (2008–2020), we divided China into main grain production areas, main marketing areas, and balanced production and marketing areas, and tested the heterogeneity of agri-ecological policy. The results are shown in regression (1) of Table 6.

**Table 6.** Regional heterogeneity test of agri-ecological policy on agricultural green technology progress.

| Variables | Regression (1) | | | Regression (2) | |
|---|---|---|---|---|---|
| | Main Production Areas | Main Sales Area | Balanced Area | Southeast Side | Northwest Side |
| $AEP^2$ | 0.158 ** | 0.196 *** | 0.288 *** | 0.153 *** | 0.159 |
| | (2.02) | (3.35) | (3.34) | (3.30) | (1.14) |
| $AEP$ | −1.068 * | −1.043 ** | −1.895 *** | −0.889 *** | −0.871 |
| | (−1.83) | (−2.52) | (−3.24) | (−2.68) | (−0.86) |
| $TRA$ | 0.038 | 0.064 | 1.540 ** | 0.068 | 0.441 |
| | (0.08) | (0.48) | (2.33) | (0.43) | (0.30) |
| $ADR$ | −0.220 | −0.050 | −0.374 | −0.317 ** | −0.002 |
| | (−0.78) | (−0.31) | (−1.40) | (−2.00) | (−0.01) |
| $MAC$ | 2.022 *** | −0.533 *** | 1.294 *** | 0.737 *** | 0.786 |
| | (5.62) | (−3.10) | (3.04) | (3.99) | (0.73) |
| $URB$ | 2.189 ** | 1.411 *** | 0.575 | 2.463 *** | −1.078 |
| | (2.29) | (3.09) | (0.51) | (4.82) | (−0.86) |
| $Cons$ | 1.146 | 1.739 ** | 3.366 *** | 0.928 | 2.163 |
| | (1.04) | (2.27) | (2.97) | (1.45) | (1.25) |
| Individual effect | Yes | Yes | Yes | Yes | Yes |
| Time effect | Yes | Yes | Yes | Yes | Yes |
| $R^2$ | 0.548 | 0.692 | 0.537 | 0.607 | 0.182 |

Note: *** indicates $p < 0.01$, ** indicates $p < 0.05$, * indicates $p < 0.1$; numbers in parentheses are t-values.

It can be seen that the effect of agri-ecological policy in the main grain producing areas, the main marketing areas and the balanced production and marketing areas all show a "U"-shaped relationship. The " U"-shaped relationship between agri-ecological policy

and agricultural green technology progress first appeared in the major food producing regions. This is the case because labor and the land factors in the main grain producing areas are richer than that in other areas. As it is the core area of grain production, China's agricultural policy and financial subsidy has continuously tilted towards the main grain producing areas. In addition, China's government has also increased investment in agricultural technology research in the main production areas, which will offset the negative effects of agri-ecological policy. The turning point of "U"-shaped agri-ecological policy in the balanced production and marketing areas appears to be latest. This is because technology intensity in the area is low and the government does not pay enough attention to agricultural development.

There are large regional differences in the natural environment and agricultural base of China. The terrain on the southeast area of China consists mostly of plains and hills, with abundant precipitation and more agricultural labor. The northwest area is dominated by deserts and plateaus, with little precipitation and a small amount of agricultural labor. Thus, we divided China into a southeast area and a northwest area according to the Hu Huanyong line, which is also cited as a sudden change line for the ecological environment. The measurement results are shown in regression (2) of Table 6. It can be seen that on the northwest side of the Hu Huanyong line, the effect of agri-ecological policy is not significant. This is because on the northwest side of the Hu Huanyong line, there are fewer people and there is poor infrastructure, which weakens the impact of agri-ecological policy.

## 5. Discussion

Agri-ecological policy is an important means for promoting agricultural green technology progress and also the basic guarantee for the green development of agriculture. Previous studies have mostly discussed the impact of agri-ecological policy on agricultural green technology progress from a static perspective. For example, some scholars found that agri-ecological policy can increase the carbon emission reduction potential of agricultural green technology progress [68]. Some scholars found that economic agri-ecological policy and administrative agri-ecological policy have a significant negative impact on agricultural technology progress [69]. However, few studies have discussed the dynamic relationship between agri-ecological policy and agricultural green technology progress. In fact, with the change of agri-ecological policy intensity, its impact on agricultural green technology progress will change. One of our contributions is to consider the dynamic impact of agri-ecological policy on agricultural green technology progress. This approach is similar to the approach in some existing studies [70,71]. Compared with these [70,71], we do not focus on the impact of agri-ecological policy on green technology progress in the manufacturing and financial sectors, but expand it to the agricultural sector. Our study verifies the "U"-shaped relationship between agri-ecological policy and agricultural green technology progress, which verifies our Hypothesis 1. It not only improves the related research of agricultural green technology progress, but also provides new ideas for promoting agricultural green development.

Previous studies have shown that human capital is closely related to agri-ecological policy and agricultural green technology progress. For example, some scholars found that agri-ecological policy has a "U"-shaped effect on the accumulation of human capital [72]. Another scholars found that human capital in terms of labor has a significant positive correlation with agricultural technology progress [61]. However, few studies have explored the relationship among the three factors. Our second contribution is to incorporate the three factors into an analytical framework to explore the impact of agricultural producer's human capital on agri-ecological policy and agricultural green technology progress. It is not only conducive to further exploring the action mechanism of agri-ecological policy on agricultural green technology progress, but also provides a new path for the green development of China's agriculture. We found that human capital can positively moderate the relationship between agri-ecological policy and agricultural green technology progress, which verifies our Hypothesis 2. At the same time, we divided the data samples into

two sides of the Hu Huanyong line and different grain production functional zones for heterogeneity analysis, which will help provide policy implications for the government to formulate different degrees of agri-ecological policy intensity and promote agricultural green technology progress.

This study also has some limitations: First, we only use provincial data to test the relationship between agri-ecological policy and agricultural green technology progress from a macro perspective. From a micro perspective, whether the prefecture-level city data support our conclusions is the focus of our next work. Secondly, this study only conducted heterogeneity analysis from the perspective of regional and food production function zone positioning. Factors such as the degree of economic development may also be reasons behind the different effects of agri-ecological policy on agricultural green technology progress. In upcoming work, we will explore the heterogeneity of agri-ecological policy on agricultural green technology progress under other conditions, so as to provide policy implications for the scientific formulation of agri-ecological policy. Finally, in this paper, we use the average years of schooling to measure the role of human capital in the impact of agri-ecological policy on the agricultural green technology progress. At the same time, the human capital of agricultural producers is not only related to their schooling level, but also to their cultural level, environmental ethics and other factors. Thus, in future research, we will consider the complex impact of agri-ecological policy on the agricultural green technology progress given different cultural levels, environmental ethics and other factors.

## 6. Conclusions

This paper uses the Super-efficiency SBM-DEA Model to measure agricultural green technology progress in 30 provinces (except Tibet) in mainland China from 2001 to 2019, and analyzes the nonlinear effects of agri-ecological policy on agricultural green technology progress using a two-way fixed-effects model, further analyzing the moderating role of agricultural producers' human capital using a moderating effects model. The conclusions are as follows:

First, there is a "U"-shaped relationship between agri-ecological policy and agricultural green technology progress. This finding still holds after four robustness tests: replacing explanatory variables, excluding some samples, adjusting the sample size, and lagging the core variables by one period. It shows that in the early stage of agri-ecological policy implementation, it hinders the agriculture green technology progress. Only after crossing the inflection point can agri-ecological policy promote the growth of agricultural green technology progress.

Second, human capital plays a positive moderating role in the effect of agri-ecological policy on agricultural green technology progress. Intermediate human capital and advanced human capital can both positively moderate the relationship between agri-ecological policy and agricultural green technology progress, while the moderating effect of primary human capital is not significant.

Finally, the "U"-shaped relationship between agri-ecological policy and agricultural green technology progress is characterized by regional heterogeneity based on differences in grain functions and differences on both sides of the Hu Huanyong line. From the perspective of each grain functions area, agri-ecological policy and agricultural green technology progress both show a "U"-shaped relationship, among which the "U"-shaped curve inflection point appears earliest in the main production area, followed by the main marketing area, and in the production and marketing balance area appears the latest. In addition, this "U"-shaped relationship is reflected on the southeast side of the Hu Huanyong line, while for the northwest side of the Hu Huanyong line, it is not obvious.

**Author Contributions:** G.M. conceived, designed, and conducted the study. M.L. revised the manuscript. T.J. and Y.L. were involved in the analysis interpretation of data and funded the study. All authors have read and agreed to the published version of the manuscript.

**Funding:** This research was funded by the National Social Science Foundation of China key project (No. 21AJY013), the International Cooperation and Exchange project of the National Natural Science Foundation of China (No. 71961147002), Guangxi Philosophy and Social Science Project (No. 22FJY021).

**Institutional Review Board Statement:** Not applicable.

**Informed Consent Statement:** Not applicable.

**Data Availability Statement:** All the data were obtained from the China Statistical Yearbook, China Rural Statistical Yearbook, Statistical Data of National Rural Economy, Annual Statistical Report of China's Rural Operation and Management, Annual Statistical Report of China's Rural Policy and Reform, Statistical Yearbook of China Population and Employment and China Agricultural Products Trade Development Report (2001–2020) and are available on request from the corresponding author.

**Acknowledgments:** Authors acknowledge the support provided by their respective institutions.

**Conflicts of Interest:** The authors declare no conflict of interest.

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
