# Peer review of "Agri-Ecological Policy, Human Capital and Agricultural Green Technology Progress"

_agriculture, doi:10.3390/agriculture13050941_

Round 1

Reviewer 1 Report

When the article is examined, it should be noted that there are limited weaknesses that need to be developed scientifically. This will be a very positive contribution if the authors work on the below issues and improve the article.  You can see detailn in my report.

Author Response

Dear Reviewers,

Thank you so much for allowing us to revise and resubmit our paper. My co-authors Guoqun Ma, Minjuan Li, Yuxi Luo and self are pleased to submit a revised version of the paper titled " Agri-ecological policy, Human Capital and Agricultural Green Technology Progress" for consideration for possible publication in the section of Article of Agriculture. This submission includes a manuscript, the response and the highlights. We have made every possible effort to use the feedback to improve the manuscript. We believe that the guidance has resulted in a substantially improved paper. We would like to thank you once again for your support and encouragement.

We look forward to your feedback on this revised submission.

Sincerely,

Guoqun Ma, Guangxi Normal University, Guilin, China

Minjuan Li, Guangxi Normal University, Guilin, China

Yuxi Luo, Guangxi Normal University, Guilin, China

Tuanbiao Jiang, Guangxi Normal University, Guilin, China

Reviewer 2 Report

1-In the section "Results of Agri-ecological Policy on agricultural green technology progress." Some of the determined coefficients are in the range of negative numbers and sometimes in different ranges; it is necessary to add explanations in this regard to the text of the article. Especially about "Coefficient" and "t-value".
2-It is better to test the relationship between agricultural-ecological policies and agricultural green technology with average human capital and advanced human capital with cultural issues and environmental ethics of the society.
3-
In the discussion section, I was expecting a review of other models in "green agriculture" and political and cultural issues, as well as pointing out the limitations of the research, which seems insufficient.
4-
One of the things that need to be emphasized is the "complex effect". Which is not mentioned directly and only a separate impact has been investigated.

Author Response

(The authors gave the same response as above.)

Reviewer 3 Report

The paper is a good attempt to link agroecological policies with adoption of green technology at sub-national level with human capital as mediating factor. The paper used good dataset and methodology. Please introduce the concepts and hypothesis in a flow chart form so that it will be clearly visible and appealing with theoretical background.

Introduce some technology diffusion models to explain the factors which influence adoption of green technology and institutional support required.

For ready reference for adoption of soil health card, which is a green technology is available.

Reddy, A. A. (2019). The soil health card Scheme in India: Lessons learned and challenges for replication in other developing countries. Journal of Natural Resources Policy Research, 9(2), 124-156 and also conditions for adoption of organic agriculture is  can be seen in Reddy, A. A., Melts, I., Mohan, G., Rani, C. R., Pawar, V., Singh, V., ... & Bhattarai, M. (2022). Economic Impact of Organic Agriculture: Evidence from a Pan-India Survey. Sustainability, 14(22), 15057.

Overall, the paper is a good addition for the literature, but the flow of the paper and definitions/concepts of the terms used needs to be improved for more readability. Concepts needs to be clearly explained before the discussion.

The below are some specific suggestions.

Clearly define ‘Agricultural Green Technology progress’

What is ‘fertilizer alternative technology’ clearly define before introducing the term first time.  What are the components of agroecological policies?

It is not clearly explained what is ‘moderating role’, may be you can replace ‘moderating’ role with mediating or accelerating role?

This will reduce the 102 inclination of agricultural producers on production decisions, negatively affecting the ag-103 agricultural green technology progress (line 102, page 3) not clear.

Can you use spill over effect rather than siphoning off, (line 148, page 4).

Number of pesticides used and number of agricultural films used (page 5), check the units, whether they are right?

Table 5 what is the dependent variable and units(each table needs to be self-explanatory).

Author Response

(The authors gave the same response as above.)

Round 2

Reviewer 2 Report

The subject of the article is not a very special subject, and accordingly, as much as the quality and the subject of the research, the proposed and required correction items and comments have been addressed.

Author Response

Dear Reviewer,

We are grateful to your valuable comments and favorable decision. We will keep trying to make our future research to meet a higher expectation.

Thanks very much for your support and ecouragement.

With regards,

Guoqun Ma, Minjuan Li, Yuxi Luo, Tuanbiao Jiang 

Reviewer 3 Report

Now the paper is acceptable for publication, The paper examined the adoption of green technologies stimulated by policies and mediated through human development indicators. The methodology and data are appropriate. Now China needs to shift to green technologies, hence the paper is timely and addressed important issue of role of human development in this process. 

Author Response

Dear Reviewer,

We are grateful to your valuable comments and favorable decision. Indeed, China comes to a point of shifting to green technology, more issues such as human capital development are worth to explore. With your encouragement, we look forward to making further contributions into this area.  

Thanks very much for your support.

With regards,

Guoqun Ma, Minjuan Li, Yuxi Luo, Tuanbiao Jiang